# Dietary Intake, Eating Behavior, Physical Activity, and Quality of Life in Infertile Women with PCOS and Obesity Compared with Non-PCOS Obese Controls

**DOI:** 10.3390/nu13103526

**Published:** 2021-10-08

**Authors:** Zheng Wang, Henk Groen, Astrid E. P. Cantineau, Tessa M. van Elten, Matty D. A. Karsten, Anne M. van Oers, Ben W. J. Mol, Tessa J. Roseboom, Annemieke Hoek

**Affiliations:** 1Department of Obstetrics and Gynecology, University of Groningen, University Medical Centre Groningen, 9700 RB Groningen, The Netherlands; wangzhengmedicine@gmail.com (Z.W.); a.e.p.cantineau@umcg.nl (A.E.P.C.); a.m.van.oers@umcg.nl (A.M.v.O.); 2Department of Epidemiology, University of Groningen, University Medical Centre Groningen, 9700 RB Groningen, The Netherlands; h.groen01@umcg.nl; 3Department of Public and Occupational Health, Amsterdam University Medical Centre, Vrije Universiteit Amsterdam, De Boelelaan 1117, 1105 AZ Amsterdam, The Netherlands; tessavanelten@gmail.com; 4Department of Clinical Epidemiology, University of Amsterdam, Amsterdam University Medical Centre, Biostatistics and Bioinformatics, Meibergdreef 9, 1105 AZ Amsterdam, The Netherlands; m.d.a.karsten@umcg.nl (M.D.A.K.); t.j.roseboom@amsterdamumc.nl (T.J.R.); 5Department of Obstetrics and Gynecology, University of Amsterdam, Amsterdam University Medical Centre, Meibergdreef 9, 1105 AZ Amsterdam, The Netherlands; 6Amsterdam Public Health Research Institute, 1105 AZ Amsterdam, The Netherlands; 7Amsterdam Reproduction and Development, 1105 AZ Amsterdam, The Netherlands; 8Department of Obstetrics and Gynecology, Monash University, Melbourne 3800, Australia; ben.mol@monash.edu

**Keywords:** PCOS, obesity, dietary intake, physical activity, eating behavior, quality of life

## Abstract

To personalize lifestyle advice for women with polycystic ovary syndrome (PCOS) and obesity, detailed information regarding dietary intake, eating behavior, physical activity levels, and quality of life (QoL) may be useful. We aimed to investigate in a post-hoc cross-sectional analysis within a large multicenter randomized controlled trial in women with infertility whether there are significant differences in dietary intake (vegetables, fruits, sugary drinks, alcoholic beverages, savory snacks, and sweet snacks); eating behavior (emotional eating, external eating, and restricted eating); physical activity; and QoL between women with PCOS and obesity and non-PCOS obese controls. Participants were asked to complete the food frequency questionnaire (FFQ), the Dutch Eating Behavior Questionnaire (DEBQ), the Short QUestionnaire to ASsess Health-enhancing physical activity (SQUASH), and the 36-item Short Form Health Survey (SF-36) at study entry (PCOS: *n* = 170; non-PCOS: *n* = 321, mean BMI: 36). Linear and binary (multinomial) logistic regressions were used, and the analyses were adjusted for age, waist–hip circumference ratio, and homeostasis model assessment of insulin resistance (HOMA-IR). No statistically significant differences in dietary intake or physical activity were observed between the two groups. The overall score of emotional eating was 34.6 ± 11.2 in the PCOS group and 34.1 ± 11.3 in the non-PCOS group (*p* = 0.11). QoL scores (physical and mental) did not differ between PCOS and non-PCOS women. These findings suggest that infertile women with PCOS and obesity and infertile non-PCOS obese controls do not have different dietary habits and have similar mental and physical QoL.

## 1. Introduction

Polycystic ovary syndrome (PCOS) is the most common endocrinopathy with reproductive, metabolic, and psychological features, affecting 5–25% women of reproductive age worldwide [1]. The main clinical features include menstrual irregularities, subfertility, hyperandrogenism, and hirsutism [1]. Up to 60% of women with PCOS are overweight or obese [2]. It is hypothesized that there is a bidirectional interaction between obesity and PCOS [3]. The pathophysiological mechanisms behind this bidirectional interaction in women with PCOS might be driven by a decreased postprandial thermogenesis and an impaired regulation of gut hormones that control appetite, leading to weight gain [4,5]. Obesity, in turn, will exacerbate the symptoms of PCOS, with BMI being the strongest correlate of PCOS status, with every BMI point increment increasing the risk of PCOS by 9% [6]. In addition to obesity, women with PCOS have an increased risk of insulin resistance and hyperinsulinemia [7]. Elevated insulin levels in women with PCOS stimulate ovarian androgen synthesis, and elevated androgen levels disrupt glucose and insulin regulation [7,8]. As a result, the presence of insulin resistance in women with PCOS increases the risk of chronic diseases, such as diabetes mellitus, cardiovascular diseases, metabolic syndrome, and even breast and endometrial cancers [9].

Studies have assessed the balance between caloric intake and caloric expenditure in women with PCOS but have shown inconsistent results [10,11,12,13,14]. The relatively small sample size, heterogeneity of the study populations, and the different BMI categories make it difficult to compare studies. While some studies reported no differences in the daily energy intake between women with and without PCOS [10,11,12,13], others demonstrated that women with PCOS had a higher energy intake and a better quality of food (lower glycemic index and lower saturated fat) [14]. A clear limitation in the latter study is the difference in BMI between the PCOS and non-PCOS groups (29.3 vs. 25.6). Therefore, their significant findings in calorie intake and food quality may be due to the BMI itself rather than PCOS. Moreover, there is little research investigating the eating behaviors (i.e., emotional eating, external eating, and restricted eating) in women with PCOS. Particularly, external eating and emotional eating behaviors in our “obesogenic society” could influence the dietary intake, leading to higher chances of weight gain [15,16]. On the caloric expenditure side of the balance with caloric intake, it has been shown that women with PCOS have an increased sitting time, although the difference might be due to higher BMI in the PCOS group [14]. 

According to the recent international evidence-based guidelines [17], lifestyle interventions should be recommended in all women with PCOS and overweightness or obesity to reduce their weight, central obesity, and insulin resistance. In particular, it is advocated that dietary approaches should achieve specific goals, such as improving insulin resistance and metabolic and reproductive outcomes. These goals could be met by low-calorie diets, limiting sugar and refined carbohydrate intakes, and reducing saturated and trans fatty acids [9,18,19]. To reach these goals, lifestyle coaching is needed; however, it has been reported that women with PCOS do not get adequate lifestyle advice from clinicians [20]. According to a recent survey in the United Kingdom, less than 50% of dietitians provided specific information on an optimal dietary intake to women with PCOS [21]. 

To adapt lifestyle advice to this group of women, information regarding dietary intake, eating behavior, physical activity levels, and quality of life (QoL) may be useful. We performed a large multicenter randomized controlled trial (RCT) to investigate the effect of lifestyle intervention on the live birth rate in women with obesity and infertility [22]. In total, 577 women with or without PCOS were included. In view of the contradictory results of the abovementioned studies and before introducing standard lifestyle interventions in women with PCOS and obesity, we aim to investigate in a post-hoc analysis within this RCT whether there are significant differences in the dietary intake (i.e., vegetables, fruits, sugary drinks, alcoholic beverages, savory snacks, and sweet snacks); eating behavior; moderate-to-vigorous physical activity; and QoL between women with PCOS and obesity and non-PCOS obese controls.

## 2. Materials and Methods

### 2.1. Study Population

This is a post-hoc cross-sectional analysis of the LIFEstyle study [22,23]. The LIFEstyle study was a multicenter RCT (Netherlands Trial Registry number: NTR 1530) approved by the Institutional Review Board of the University Medical Center Groningen (no.: 2008/284). In the original RCT, between 2009 and 2012, a total of 577 women with or without PCOS between 18 and 39 years old and with a BMI ≥ 29 were randomly assigned to a six-month lifestyle intervention, followed by 18 months of infertility treatment or to immediate infertility treatment for 24 months. The baseline measures from the RCT were used in the current analysis. Women were diagnosed as PCOS in cases where two of the three Rotterdam criteria were met [24]: oligo-ovulation or anovulation, clinical manifestations of hyperandrogenism and/or hyperandrogenemia, or ovarian polycystic changes. The control group consisted of non-PCOS obese women (anovulatory and ovulatory). 

### 2.2. Clinical and Laboratory Measurements

Body weight (kg), height (cm), waist and hip circumference (cm), and blood pressure (mmHg) were measured by research nurses who were blinded to the treatment assignment. Fasting glucose, insulin, triglycerides, total cholesterol, low-density lipoprotein cholesterol (LDL-C), high-density lipoprotein cholesterol (HDL-C), and high-sensitivity C-reactive protein (hs-CRP) were measured. A homeostasis model assessment of insulin resistance (HOMA-IR) was calculated as the fasting insulin concentration (μU/mL) multiplied by the fasting glucose concentration (mmoL/L) divided by 22.5. The measurement method, as well as intra- and inter-assay variations of these outcomes, have been described elaborately [25].

### 2.3. Diet, Eating Behavior, Physical Activity, and QoL

All participants were asked to complete the food frequency questionnaire (FFQ) [26], the Short QUestionnaire to ASsess Health-enhancing physical activity (SQUASH) [27], the Dutch Eating Behavior Questionnaire (DEBQ) [28], and the 36-item Short Form Health Survey (SF-36) [29] at the study entry. Participants who provided FFQ and/or SQUASH at the baseline constituted the population for the current analysis. Women who were pregnant at the time of filling out the questionnaire were excluded, since pregnancy may result in lifestyle and dietary changes. 

The FFQ consisted of two parts. The first part was a standardized questionnaire consisting of questions about the type of cooking fats; type of bread; frequency of breakfast use; frequency of consumption; and portion sizes of vegetables, fruits, and fruit juices. The questions on the intake of fruits, fruit juices, and cooked vegetables were validated against two 24-h recalls [26]. The additional part consisted of questions about their snack intake, consumption of sugar-containing and alcoholic beverages, and the use of cream and/or sugar in coffee and tea. The frequency of weekly or monthly consumption was asked. Portion sizes for all foods and food groups were recorded in the usual household measures (e.g., glass or handful). Thus, we derived quantitative information regarding the intake of vegetables (grams/day), fruits (grams/day), sugary drinks (glasses/day), alcoholic beverages (glasses/day), savory snacks (handful/week), and sweet snacks (portion/week). The number of sugary drinks (<0.5, 0.5–1, and >1 glasses/day); savory snacks (<2, 2–10, and >10 handful/week); and sweet snacks (<2, 2–10, and >10 handful/week) were arbitrarily divided into three levels based on the percentile ranges. Physical activity was quantified by collecting information about commuting activities, leisure time activities, household activities, and activities at work and school. The results were presented as the following outcome measures: moderate-to-vigorous leisure time physical activity (yes/or), moderate-to-vigorous commuting activities (yes/no), and total moderate-to-vigorous physical activity (minutes/week). We also arbitrarily divided the total moderate-to-vigorous physical activity into three levels based on the percentile ranges (<200, 200–700, and >700 min/week). The effect of lifestyle intervention on diet and physical activity during and after the study period has been published, and the measurements of diet and physical activity have been elaborately described [30,31]. 

The DEBQ is a 33-item instrument comprised of three subscales that measure emotional, external, and restrained eating, which was developed by van Strien in 1986 [28]. The emotional eating subscale (13 items) measures the tendency to use food to cope with psychological problems and/or to alleviate distress. The external eating subscale (10 items) assesses the frequency of eating in response to external stimuli (e.g., the appearance and smell of food). The restrained eating subscale (10 items) assesses the frequency of restrictive conscious behaviors during eating. Each item was rated on a five-point scale ranging from 1 (never), 2 (seldom), 3 (sometimes), and 4 (often) to 5 (very often), with higher scores indicating more severe eating disorders. The SF-36 is a general health-related QoL measure consisting of 36 items [29]. This questionnaire consists of a Physical Component Score and a Mental Component Score, in which higher scores represents a better QoL. 

### 2.4. Statistical Analysis

SPSS Statistics 25 (IBM, Chicago, IL, USA) was used to perform the analyses. Normality testing was performed using histograms combined with the Kolmogorov–Smirnov test. As appropriate, the data were presented as the mean ± SD or median with an interquartile range (IQR) for the continuous variables. The categorical variables are presented as counts (percentage). We compared the baseline characteristics between women with and without PCOS using the chi-square test for categorical variables and Student’s *t*-test or the Mann–Whitney *U* Test for continuous variables. For comparison of the dietary intake, eating behavior, physical activity, and QoL between women with and without PCOS, linear regression and binary (multinomial) logistic regression were used as appropriate. We first calculated the crude association, expressed as B or OR. We subsequently adjusted for variables that showed statistically significant differences between the groups. With respect to variables showing big differences between the crude and adjusted models (e.g., more than 20% change in B or OR), we performed further analyses adjusting for confounders one at a time to determine which one influences the association the most.

## 3. Results

The data of 574 women included in the LIFEstyle study were available (three women withdrew informed consent). Two women had unknown ovulatory statuses and could not be included in this analysis, leaving 201 women who were diagnosed as PCOS at the study entry and 371 non-PCOS women. A total of 178 women in the PCOS group and 330 in the non-PCOS group completed the FFQ and/or SQUASH. The exclusion of women who were pregnant at the time of the questionnaire led to 170 PCOS and 321 non-PCOS women in the current analysis (Figure 1). Table 1 shows the baseline characteristics of the participants. Women with PCOS were younger than non-PCOS women (28.0 ± 4.2 vs. 30.8 ± 4.4 years, *p* < 0.001). Although there was no statistically significant difference in the BMI (mean BMI: 36 in both groups), the waist–hip circumference ratio was slightly higher in women with PCOS (0.87 ± 0.07 vs. 0.86 ± 0.06, *p* = 0.02). The fasting insulin levels and HOMA-IR were higher in women with PCOS. The other characteristics were not statistically different between the groups. Thus, age, waist–hip circumference ratio, and HOMA-IR (since insulin and HOMA-IR are highly correlated, we only included HOMA-IR) were included in the adjusted regression models to investigate the differences in diet, eating behavior, physical activity, and QoL.

A comparison of the diet, eating behavior, physical activity, and QoL between the PCOS group and the non-PCOS group is shown in Table 2. Women with PCOS had an intake of 107 (78.6; 157) gram vegetables per day, which was significantly less than non-PCOS women (129 (85.7; 179) gram). However, after adjusting for age, waist–hip circumference ratio, and HOMA-IR, the statistically significant difference disappeared (mean difference: −8.31 g, 95% CI: −25.0 to 8.36, *p* = 0.33). The analyses adjusting for age, waist–hip circumference ratio, and HOMA-IR one at a time (Appendix A) showed that the change of the mean difference was mainly driven by age. No statistically significant differences in the intake of fruits, sugary drinks, alcoholic beverages, savory snacks, and sweet snacks were observed between the two groups. The overall score of emotional eating was 34.6 ± 11.2 in the PCOS group and 34.1 ± 11.3 in the non-PCOS group (mean difference: 2.03, 95% CI: −0.45 to 4.51, *p* = 0.11). The overall scores of external eating and restricted eating in both groups were not statistically significantly different. No statistically significant differences in the total moderate-to-vigorous physical activity were observed between the groups, as well as the proportion of leisure time moderate-to-vigorous physical activity (yes/no) or commuting moderate-to-vigorous physical activity (yes/no). As for QoL, both the physical QoL (mean difference: −1.26, 95% CI: −3.29 to 0.76, *p* = 0.22) and mental QoL (mean difference: 0.45, 95% CI: −1.72 to 2.62, *p* = 0.69) were similar between the groups.

## 4. Discussion

We investigated whether there were differences in dietary intake across various food groups, eating behaviors, physical activities, and QoL between women with PCOS and obesity and non-PCOS obese controls in a post-hoc cross-sectional analysis. We found no significant differences in the dietary intake (vegetables, fruits, sugary drinks, alcoholic beverages, savory snacks, and sweet snacks); eating behavior (emotional eating, external eating, and restricted eating); total, leisure time, and commuting moderate-to-vigorous physical activities; and QoL between women with PCOS and obesity and non-PCOS obese controls.

Our findings with respect to differences in the dietary intake between infertile women with obesity with and without PCOS are in alignment with previous small studies, although the types of food reported in different studies vary [11,13], and the BMI differed in the published study between the PCOS and non-PCOS participants (31.5 vs. 28.0) [13]. For instance, no significant differences were found in the intake of fruits, vegetables, and sugary drinks between women with obesity with and without PCOS in both Altieri’s [11] and our study. Our study adds to the current literature, since our participants had similar high BMI. In addition to the dietary intake, our analysis shows a similar total physical activity between women with PCOS and non-PCOS women, which is in line with previous studies [13,14]. Evidence regarding the eating behaviors between women with PCOS and non-PCOS women is scarce. One relevant study was performed by Larsson et al. [32], who reported that there were no significant differences in the eating behaviors between the PCOS and controls assessed by the 21-item Three-Factor Eating Questionnaire (PCOS: *n* = 72; non-PCOS: *n* = 30). However, there was a statistically significant difference in the BMI between their PCOS and non-PCOS participants (28.5 vs. 24.6). Our findings with respect to eating behaviors advance the knowledge on this topic by showing that PCOS does not seem to be associated with specific eating behaviors, even when the BMI of the control group is comparable to the PCOS group. Moreover, our results are based on a larger, well-defined cohort compared to Larsson’s study using the DEBQ, a tool for assessing eating behaviors similar to the 21-item Three-Factor Eating Questionnaire.

In addition to the dietary intake and physical activity level in women with PCOS, mental QoL is another important parameter to be considered before giving advice on a healthy lifestyle. A poor mental QoL and depression have been reported in women with PCOS [33,34], and women with a poorer mental QoL have more difficulties following a lifestyle intervention [35,36]. Our results showed no statistically significant differences in physical and mental QoL scores between women with PCOS and obesity and non-PCOS obese controls, contrary to a recent systematic review [37], which reported that PCOS has negative effects on the mental QoL. However, the author of that systematic review argued that the negative effect on the mental QoL might be due to obesity itself, given that obesity contributes substantially to mental problems in women with PCOS [38]. Our results, showing no statistically significant differences in the physical and mental QoL scores between the groups, seem to further validate this conclusion. More studies where women with similar BMI but who differ with respect to PCOS status are compared are needed to verify this conclusion.

The strengths of our study were the sample size, the well-defined population of PCOS and non-PCOS women, the fact that both groups have similar BMI, and the prospective data collection. However, some limitations should be mentioned. The most notable limitation in the current analysis was the difference in age between the two groups, which may have influenced our findings. This was exemplified by the disappearance of statistically significant findings in the intake of vegetables in the models adjusted for age. Nonetheless, the age difference was expected, since all included women were infertile. In general, women with PCOS go to the hospital for consultation or treatment due to irregular periods or menopause earlier than for other causes of infertility. Another limitation was the use of self-reported questionnaires; this might lead to the over-reporting of healthy behavior and underreporting of unhealthy behavior [39,40]. However, it is unlikely that this might have influenced the results, because both groups were obese and had similar BMI. The FFQ only investigated certain types of food products, so we were not able to analyze the differences regarding nutrients, such as carbohydrates, dietary fiber, fat, protein, minerals, vitamins, or micronutrients, and this limited the comparisons of our findings with previous studies [10,11]. Due to the original study design of the RCT, the average calorie intake per day was only collected and calculated in women who were allocated to the intervention arm and, thus, could not be reported here. Furthermore, in the current analysis, we aimed to exclude women with apparent or known eating disorders, but this was not supported by formal diagnoses and may have resulted in a bias in our findings. Finally, all the women included in the current analysis were infertile, limiting the generalizability of our results to fertile women.

Exploring whether women with PCOS and obesity have different dietary habits, physical activity, and QoL compared with non-PCOS obese controls is important, since it could provide further clinical guidance with respect to what kind of adaptations might be needed in lifestyle intervention programs for these women, with more-or-less emphasis on diet or physical activity. Our findings in the current analysis suggest that the presence of PCOS in an infertile woman may not require different lifestyle advice than in obese infertile women in general. Whether this also implies that the effectiveness of lifestyle interventions for women with obesity is not influenced by their PCOS status requires further investigation.

## 5. Conclusions

In summary, infertile women with PCOS and obesity do not appear to have different dietary intakes, eating behaviors, physical activity, and QoL compared with infertile non-PCOS obese controls.

## Figures and Tables

**Figure 1 nutrients-13-03526-f001:**
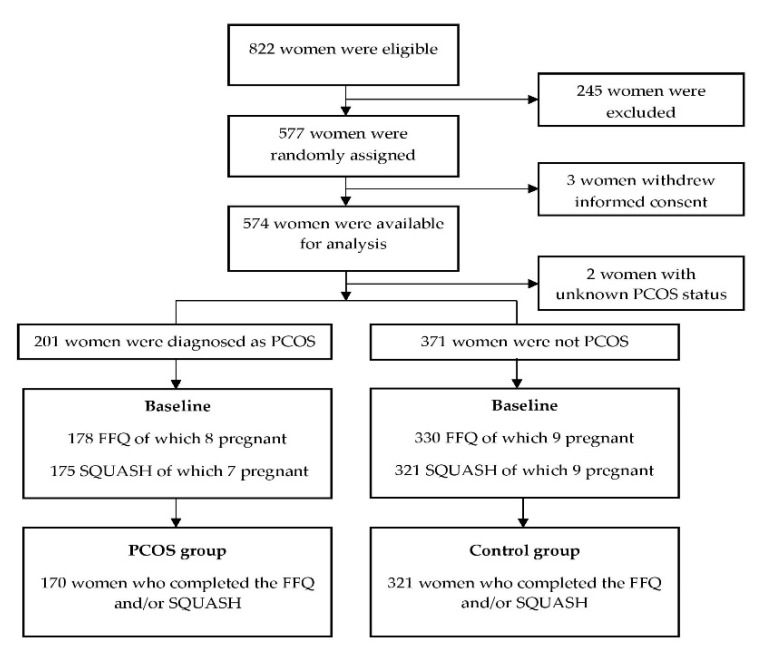
Flow chart of the participants who completed the FFQ and/or SQUASH at the baseline. PCOS: polycystic ovary syndrome, FFQ: food frequency questionnaire, and SQUASH: Short QUestionnaire to ASsess Health-enhancing physical activity.

**Table 1 nutrients-13-03526-t001:** Baseline characteristics of the participants who completed the FFQ and/or SQUASH at the baseline.

	PCOS (*n* = 170)	non-PCOS (*n* = 321)	*p*-Value
Age (y)	28.0 ± 4.2	30.8 ± 4.4	<0.001
Western European Ethnicity	153 (90%)	287 (89.4%)	0.84
Education			0.79
Primary school	6 (3.8%)	12 (3.9%)	
Secondary education	41 (25.6%)	71 (26.8%)	
Intermediate Vocational Education	79 (49.4%)	149 974.8%)	
Higher Vocational Education and University	34 (21.3%)	78 (25.2%)	
Current smoker	43 (25.6%)	73 (23.0%)	0.52
Anthropometrics			
Weight (kg)	104.0 ± 12.7	103.2 ± 13.0	0.54
BMI	36.0 ± 3.5	36.0 ± 3.3	0.93
Waist circumference (cm)	108.7 ± 9.1	107.6 ± 9.3	0.21
Hip circumference (cm)	124.6 ± 9.0	125.4 ± 8.7	0.33
Waist-hip circumference ratio	0.87 ± 0.07	0.86 ± 0.06	0.02
Blood pressure			
Systolic blood pressure (mmHg)	126.2 ± 13.6	127.7 ± 14.1	0.28
Diastolic blood pressure (mmHg)	80.0 ± 8.9	80.2 ± 9.4	0.74
Biochemical measures			
hs-CRP (mg/L)	4.6 (2.2; 8.7)	4.0 (1.9; 7.0)	0.06
Triglycerides (mmoL/L)	1.1 (0.8; 1.6)	1.1 (0.8; 1.5)	0.76
Total cholesterol (mmoL/L)	4.7 ± 0.9	4.8 ± 0.9	0.62
HDL-C (mmoL/L)	1.1 ± 0.2	1.2 ± 0.3	0.30
LDL-C (mmoL/L)	3.1 ± 0.8	3.1 ± 0.9	0.81
Fasting glucose (mmoL/L)	5.4 ± 0.9	5.4 ± 0.6	0.47
Fasting insulin (pmoL/L)	116.5 ± 60.5	92.4 ± 51.1	<0.001
HOMA-IR	4.1 ± 2.7	3.2 ± 1.9	<0.001
Metabolic syndrome	80 (59.3%)	148 (54.0%)	0.32

PCOS: polycystic ovary syndrome, HDL-C: high-density lipoprotein cholesterol, LDL-C: low-density lipoprotein cholesterol, hs-CRP: high-sensitivity C-reactive protein, and HOMA-IR: homeostasis model assessment of insulin resistance. Data are reported as the mean ± SD, median (IQR), or counts (percentage).

**Table 2 nutrients-13-03526-t002:** Comparison of the diet, eating behavior, physical activity, and quality of life between the PCOS group and the non-PCOS group.

	PCOS (*n* = 170)	Non-PCOS (*n* = 321)	Crude B or OR (95% CI)	*p*-Value	Adjusted * B or OR (95% CI)	Adjusted * *p*-Value
Diet						
Vegetable intake (g/day)	107 (78.6; 157)	129 (85.7; 179)	−17.4 (−31.7 to −3.19)	0.02	−8.31 (−25.0 to 8.36)	0.33
Fruit intake (g/day)	100 (57.1; 143)	85.7 (42.9; 142.9)	8.61 (−6.19 to 23.4)	0.25	15.4 (−1.63 to 32.4)	0.08
Sugary drinks (glasses/day)						
<0.5	49 (32.2%)	114 (40.4%)	ref	ref	ref	Ref
0.5–1	25 (16.4%)	51 (18.1%)	1.14 (0.64 to 2.05)	0.66	0.95 (0.49 to 1.86)	0.88
>1	78 (51.3%)	117 (41.5%)	1.55 (1.00 to 2.41)	0.05	0.72 (0.42 to 1.23)	0.23
Alcoholic beverages						
No	110 (68.3%)	182 (61.3%)	ref	ref	ref	ref
Yes	51 (31.7%)	115 (38.7%)	0.73 (0.49 to 1.10)	0.14	0.70 (0.43 to 1.13)	0.15
Savory snacks (handful/week)						
<2	67 (41.6%)	121 (40.5%)	ref	ref	ref	ref
2–10	53 (32.9%)	104 (34.8%)	0.92 (0.59 to 1.44)	0.72	0.91 (0.54 to 1.54)	0.72
>10	41 (25.5%)	74 (24.7%)	1.00 (0.62 to 1.62)	1.00	0.88 (0.49 to 1.57)	0.67
Sweet snacks (portion/week)						
<2	56 (35.2%)	103 (34.3%)	ref	ref	ref	ref
2–10	77 (48.4%)	153 (51.0%)	0.93 (0.61 to 1.42)	0.72	0.94 (0.57 to 1.55)	0.80
>10	26 (16.4%)	44 (14.7%)	1.09 (0.61 to 1.95)	0.78	0.95 (0.47 to 1.93)	0.89
Eating behavior						
Emotional eating overall sore	34.6 ± 11.2	34.1 ± 11.3	0.49 (−1.64 to 2.61)	0.65	2.03 (−0.45 to 4.51)	0.11
External eating overall score	27.7 ± 6.3	27.5 ± 5.8	0.23 (−0.90 to 1.35)	0.69	0.26 (−1.05 to 1.57)	0.70
Restricted eating overall score	32.3 ± 5.9	31.7 ± 6.1	0.61 (−0.53 to 1.75)	0.30	0.81 (−0.51 to 2.13)	0.23
Physical activity						
Total moderate-to-vigorous physical activity (minute/week)						
<200	54 (32.1%)	108 (34.6%)	ref	ref	ref	ref
200–700	59 (34.1%)	98 (31.4%)	1.20 (0.76 to 1.91)	0.43	1.44 (0.84 to 2.49)	0.19
>700	55 (32.7%)	106 (34.0%)	1.04 (0.65 to 1.65)	0.88	0.92 (0.53 to 1.58)	0.75
Leisure time moderate-to-vigorous physical activity						
No	40 (23.8%)	61 (19.6%)	ref	ref	ref	ref
Yes	128 (76.2%)	251 (80.4%)	0.78 (0.50 to 1.22)	0.28	0.92 (0.53 to 1.60)	0.77
Commuting moderate-to-vigorous physical activity						
No	105 (62.5%)	218 (69.9%)	ref	ref	ref	ref
Yes	63 (37.5%)	94 (30.1%)	1.39 (0.94 to 2.07)	0.10	1.29 (0.81 to 2.06)	0.28
Quality of life						
Physical Component Score	49.1 ± 9.5	50.2 ± 9.1	−1.04 (−2.85 to 0.77)	0.26	−1.26 (−3.29 to 0.76)	0.22
Mental Component Score	50.2 ± 8.5	49.4 ± 10.4	0.80 (−1.12 to 2.71)	0.41	0.45 (−1.72 to 2.62)	0.69

Data are reported as the median (IQR) or counts (percentage). * Adjusted for age, waist–hip circumference ratio, and HOMA-IR.

## Data Availability

The data presented in this study are available on request from the corresponding author. The data are not publicly available due to privacy.

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
