# Peer review of "Dietary Intake, Eating Behavior, Physical Activity, and Quality of Life in Infertile Women with PCOS and Obesity Compared with Non-PCOS Obese Controls"

_nutrients, 2021, doi:10.3390/nu13103526_

Round 1

Reviewer 1 Report

There are potential typos and mistakes such as follows.

L30, L105: 36-Item => 36-item like L225 (21-item)?

L31, L85, L169, L216, L227, etc. : m2 => 2 should be in upper small letter like one in Table 1. 

L123: Physical activity were =>  Physical activity was or Physical activities were.

L137, L156: e.g. => e.g., like other places.

L150: Student’s T-test => Student’s t-test

L237: Where is the verb of " Our results"?

L241: om => on; might due to => might be due to?

 L256: Please add some conjunction such as and between questionnaires, and this.

Author Response

Authors’ reply: Thank you for taking the time to read and evaluate our manuscript and for providing us with your comments and suggestions. We apologize for typos and errors. The contents of the manuscript have been carefully checked and revised.

Reviewer 2 Report

General comment

Thank you for giving me the possibility to comment on your paper. It is necessary to revise the wording and spelling of the item as there are several spelling and punctuation errors.

Specific comments

Materials and methods

Should be explain the criteria of food selection in the questionnaire and why other food groups were not included.

Results

Both the text and figure 1 should be reviewed since the final number of participants in each of the groups is not sufficiently clear.

Author Response

Authors’ reply: First of all, thank you for taking the time to read and evaluate our manuscript and for providing us with relevant comments and suggestions. We apologize for potential typos and errors.

We understand the reviewer’s concern. However, the detailed information about food frequency questionnaire has been published (1,2). We pointed out in the manuscript that the detailed information could be found (in line 136). Moreover, this manuscript is one of two manuscripts that will be submitted by our group for the same issue. The second manuscript will be submitted separately and addresses the longitudinal analysis of lifestyle change (diet and physical activity) of a lifestyle intervention program between women with PCOS and obesity and controls (the same cohort as this manuscript). The detailed information of food intake will be provided in that manuscript.

Reference:

  1. Van Elten, T.M.; Van Poppel, M.N.M.; Gemke, R.; Groen, H.; Hoek, A.; Mol, B.W.; Roseboom, T.J. Cardiometabolic Health in Relation to Lifestyle and Body Weight Changes 3(-)8 Years Earlier. Nutrients 2018, 10
  2. van Elten, T.M.; Karsten, M.D.A.; Geelen, A.; van Oers, A.M.; van Poppel, M.N.M.; Groen, H.; Gemke, R.; Mol, B.W.; Mutsaerts, M.A.Q.; Roseboom, T.J., et al. Effects of a preconception lifestyle intervention in obese infertile women on diet and physical activity; A secondary analysis of a randomized controlled trial. PLoS One 2018, 13, e0206888

We apologize for the misleading of the final number of participants. The text and Figure 1 have been improved by providing detailed inclusion criteria.